# A novel genus of virulent phage targeting *Acinetobacter baumannii*: Efficacy and safety in a murine model of pulmonary infection

JiaWang Wang[1,2,3], Huan Hu[2,3], QingQing Wang[2,3], TianZhu Zhu[2,3], XiaoHan Ren[2,3], WenQian Jiang[2,3], XueMei Hou[2,3]\*, Jun Lin[2,3]\*, XiaoLing Yu[1,2,3]\*

**1** Department of Pharmacy, Mengchao Hepatobiliary Hospital of Fujian Medical University, Fuzhou, China, **2** Institute of Applied Genomics, Fuzhou University, Fuzhou, China, **3** College of Biological Science and Engineering, Fuzhou University, Fuzhou, China

\* xiaolingyu82@163.com (XY); jun@fzu.edu.cn (JL); houxuemei_1990@163.com (XH)

## Abstract

*Acinetobacter baumannii* is a notable opportunistic pathogen responsible for severe hospital-acquired infections, with multidrug-resistant strains posing significant treatment challenges. Phage therapy, which employs bacteriophages as natural bacterial antagonists, has gained renewed attention as a promising solution to combat antibiotic-resistant infections. In this study, we isolated and characterized a novel virulent phage, vB_AbaS_qsb1, which specifically lyses *A.baumannii*. Phylogenetic and genomic analyses indicate that vB_AbaS_qsb1 is the founding member of a previously unreported genus, which we propose to name *Acinibactriovirus*, with *Acinibactriovirus lysinus* as the type species. vB_AbaS_qsb1 demonstrated robust stability across diverse temperature and pH ranges, a short latent period, and no known virulence or antibiotic resistance genes within its 54,713 bp dsDNA genome. Safety assessments showed that high-dose vB_AbaS_qsb1 induced no adverse effects in mice, with histopathology confirming its safety profile. Therapeutic experiments further indicated that vB_AbaS_qsb1 provided at least 50% protection against *A.baumannii*-induced pneumonia, significantly reducing bacterial loads and inflammation markers, while maintaining high phage titers in lung tissue. This study introduces vB_AbaS_qsb1 as a promising candidate for phage therapy against *A.baumannii*, offering both innovative insights and a valuable framework for future isolation, genomic characterization, and efficacy evaluation of phages targeting antibiotic-resistant bacteria.

### Author summary

In this study, we identified and characterized vB_AbaS_qsb1, a novel bacteriophage that specifically targets *A.baumannii*, a major cause of drug-resistant

**Data availability statement:** All data supporting the findings of this study are included within public repository and its supplementary materials. All original data related to the manuscript are available on the Figshare database and can be accessed via the following DOI: 10.6084/m9.figshare.27635490 (https://figshare.com/articles/dataset/My_raw-data/27635490?file=52193141). (The NGS raw sequencing data have been deposited in the SRA databases, SRA data accession: PRJNA1170883; The complete genome sequence of the phage has been uploaded to the Bankit database of genbank, accession number: PV107591).

**Funding:** This work was supported by the Fujian Provincial Department of Science and Technology under award number 2024J011223 to XLY and 2022Y4003 to JL, Young and middle-aged talent research project of Fuzhou city under award number 2022-S-rc5 to XLY, Ministry of Science and Technology of the People's Republic of China under award number 2023YFC3304304 to JL, Fuzhou University under award number 2024T021 to HH and Fujian Provincial Major Health Research Project under award number 2022ZD01001 to JL. The funders had no role in study design, data collection and analysis, decision to publish, or preparation of the manuscript.

**Competing interests:** I have read the journal's policy and the authors of this manuscript have the following competing interests: The authors would like to declare the following patents/patent applications associated with this research: [Chinese Patent Publication Number: CN119162120A].

infections in hospitals worldwide. Genetic analysis revealed that vB_AbaS_qsb1 belongs to a previously unreported genus, which we named *Acinibactriovirus*. Through experimental testing in a mouse lung infection model, we demonstrated this phage's effectiveness in reducing bacterial load, preventing lung damage, and enhancing survival—all without adverse immune responses. Our findings underscore the potential of phage therapy as an alternative or complementary approach to traditional antibiotics, especially against multidrug-resistant pathogens. As antibiotics become less effective due to increasing bacterial resistance, phages like vB_AbaS_qsb1 may offer a promising strategy for treating infections that are no longer responsive to conventional treatments. This work broadens our understanding of bacteriophage diversity and highlights their potential role in tackling antibiotic resistance, a significant and growing public health concern.

## Introduction

*Acinetobacter baumannii* is a Gram-negative, non-motile opportunistic pathogen commonly found in aquatic and soil environments, and a leading cause of nosocomial infections. It is associated with high mortality rates, particularly in Intensive Care Unit (ICU) settings where infections such as pneumonia and sepsis can result in mortality rates as high as 54% [1,2]. This pathogen is a major contributor to lung infections, sepsis, meningitis, urinary tract infections, and wound infections [3].

In clinical practice, antibiotics are commonly used to treat *A.baumannii* infections, but the misuse of these drugs has led to the rise of multidrug-resistant (MDR) strains, including carbapenem-resistant *A.baumannii* [4]. While combination antibiotic therapy is employed to combat resistance [5], its effectiveness can be limited by antagonistic interactions and may even promote broad-spectrum resistance [6]. The high mortality rate of *A.baumannii* infections stems from its intrinsic antibiotics resistance and its remarkable capacity to acquire antibiotics resistance genes from the environment [7,8]. In response, the WHO classified carbapenem-resistant *A.baumannii* as a critical priority for antibiotic development in 2017, reaffirming its importance in 2024 [9,10].

Given the escalating challenge of antibiotic resistance, the need for novel antimicrobial treatments is pressing. Phage therapy, an ancient yet increasingly promising approach, has garnered renewed interest. Bacteriophages are viruses that specifically infect bacteria, and based on their lifecycle, they can be categorized as either virulent or temperate [11]. Due to their ability to specifically lyse bacterial pathogens without harming eukaryotic cells, virulent phages have emerged as potential therapeutic agents for treating bacterial infections [12]. *In vitro* studies have demonstrated the potent antibacterial activity of phages, showing their ability to effectively eliminate bacteria and disrupt biofilms [13], Jeremy J. Barr *et al.* [14,15] demonstrated that both single phage therapy and phage-antibiotic combination therapy can effectively alleviate infection symptoms in mice. Furthermore, *in vivo* studies of animals have confirmed the safety and efficacy of phage therapy in treating bacterial infections [16,17].

Numerous compassionate-use cases have highlighted the clinical success of phage therapy in treating various infections, including lung infections [18,19], pancreatic infections [20], osteomyelitis [21], and enteritis [22]. In this study, We isolated a novel phage of *A.baumannii* from sewage. We characterized its biological properties and genomic features. Remarkably, phylogenetic analysis revealed that vB_AbaS_qsb1 belongs to a previously unreported genus, which we have named as *Acinibactriovirus*, with vB_AbaS_qsb1 serving as the type species (*Acinibactriovirus lysinus*). This classification underscores the novelty of our work, as it introduces a new genus of phage. The phage has been preserved in the Guangdong Microbial Culture Collection Center (GDMCC, accession number GMDCC 64831-B1). Additionally, we developed a mouse model of *A.baumannii*-induced pneumonia to evaluate both the safety and antibacterial efficacy of vB_AbaS_qsb1 i*n vivo*. This model is essential for assessing the phage's ability to reduce bacterial loads, enhance survival rates, and confirm its safety profile, providing a robust foundation for potential clinical applications in combating resistant *A.baumannii* infections.

## Result

### Purification and morphology of bacteriophage vB_AbaS_qsb1

A *A.baumannii* phage, qsb1, was successfully isolated from local lake (QiShan lake, FuZhou city, FuJian provience, China) water sample, utilizing *A.baumannii* strain ioag01 as the host. Prepare a double-layer plate by mixing qsb1 and ioag01 evenly, and subsequent incubation at 37°C overnight, distinct plaques with an average diameter of approximately 4.0 ± 1.0 mm were observed, accompanied by a depolymerase halos with an approximate measurement of 7.0 ± 1.0 mm (Fig 1A). Transmission electron microscopy (TEM) analysis (Fig 1B) revealed that the phage's head dimensions were approximately 70 × 80 nm, with a tail extending about 90 nm in length and a tail diameter of 20 nm. In accordance with the International Committee on Taxonomy of Viruses (ICTV) classification and nomenclature criteria, the phage was designated as vB_AbaS_qsb1. The phage has been archived in the Guangdong Microbial Culture Collection Center (GDMCC) with the accession number GMDCC 64831-B1.

The study of the biological characteristics of qsb1 showed that the optimal phage replication efficiency was achieved at an multiplicity of infection (MOI) of 0.01 (Fig 1C). At this MOI, phage proliferation exhibited a two-log increase within 6 hours under these conditions. We analyzed the one-step growth curve of qsb1 at an MOI of 0.01. The results showed an extremely short latent period, with host lysis occurring within 10 minutes, followed by a rapid release of progeny phages. The phage population entered a plateau phase at 100 minutes, reaching a burst size of approximately 69 plaque-forming units (PFU)/cell (Fig 1D). Thermal stability analysis demonstrated that qsb1 remains stable between 10 °C and 60 °C, though the phage lost all activity at 80 °C and 100 °C (Fig 1E). pH stability testing indicated qsb1 stability across a pH range of 3–11, with significant reductions in activity observed at pH 1 and complete inactivation at pH 13 (Fig 1F). Overall, qsb1 shows robust stability under normal physiological conditions, with activity only compromised under extreme heat or pH levels.

### Genome characteristics of the phage vB_AbaS_qsb1

**Genome features of phage vB_AbaS_qsb1.** Pulsed-field gel electrophoresis (PFGE) estimated the genome size of vB_AbaS_qsb1 to be approximately 50 kbp (S1 Fig). Next-generation sequencing (NGS) revealed that the qsb1 genome comprises 54,713 base pairs (Fig 2) of double-stranded DNA (verified using RNAse A, DNAse I and S1 nuclease; S2 and S3 Figs), with a GC content of 39%. A total of 95 coding DNA sequences (CDSs) were identified, numbered from CDS_1 to CDS_95 (S1 Table). Among these, 42 CDSs were predicted to encode proteins with known functions, while the 51 were classified as hypothetical proteins with uncharacterized roles, and an additional 2 CDSs encoded tRNAs. The identified proteins are involved in key biological processes such as structural assembly, DNA replication and repair, gene regulation, host recognition, and infection mechanisms. Notably, an integrase gene (CDS_1), typically associated with lysogeny, was identified, but experimental data and lifestyle prediction analyses confirmed that qsb1 functions as a virulent phage in both

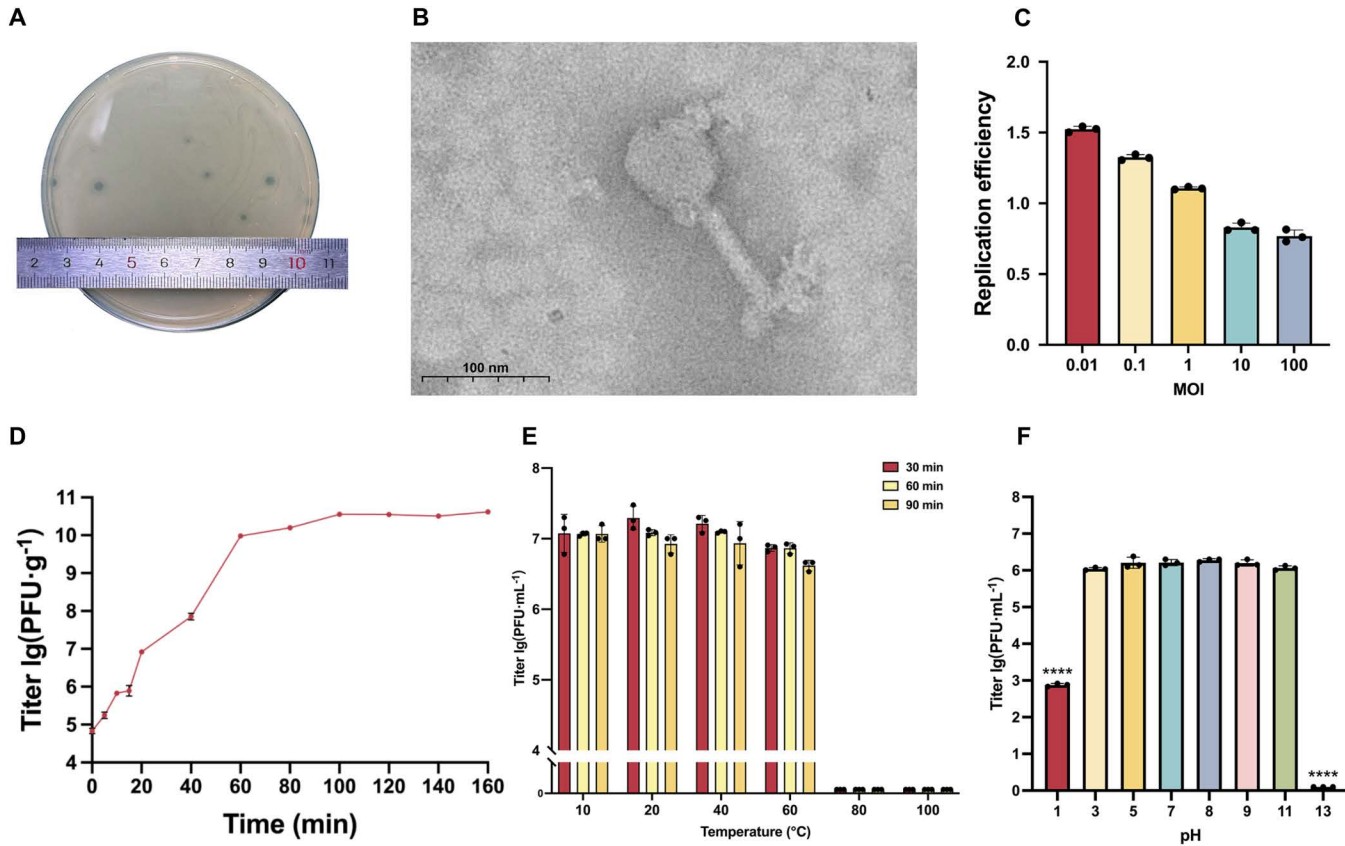

**Fig 1. Biological characteristics of bacteriophage vB_AbaS_qsb1. (A)** Plaque morphology of vB_AbaS_qsb1 on double-layer agar plates. When vB_AbaS_qsb1 and ioag01 were co-cultured in soft agar overnight at 37°C, clear transparent plaques and distinct depolymerase halos were observed. **(B)** Transmission electron microscopy (TEM) image of vB_AbaS_qsb1 morphology, negatively stained with 4% phosphotungstic acid, The scale bar corresponds to 100 nm. **(C)** MOI of vB_AbaS_qsb1, showing phage titer following infection of its host at various MOIs. MOI ranged from 0.01 to 100, with phage particle counts of $1 \times 10^6$ PFU/mL to $1 \times 10^{10}$ PFU/mL. Phage titers were measured in triplicate, and replication efficiency was calculated as the ratio of final to initial titers. **(D)** One-step growth curve of vB_AbaS_qsb1 at an MOI of 0.01. The initial bacterial concentration was $6 \times 10^8$ CFU/mL, and the phage titer reached $4.2 \times 10^{10}$ PFU/mL at the plateau phase. The burst size was calculated as the ratio of the plateau phage titer to the initial bacterial count. Data represent the mean of three independent experiments. **(E)** Temperature stability of vB_AbaS_qsb1, represented by titer variations at different temperatures across varying time periods. The initial phage titer was $1 \times 10^7$ PFU/mL. Titers were measured in triplicate, with results averaged. **(F)** pH stability of vB_AbaS_qsb1. All samples were analyzed in triplicate, and the results are expressed as mean values ±SD. The data were analyzed using two-way ANOVA. ****$p < 0.0001$, ***$p < 0.001$, **$p < 0.01$, *$p < 0.05$.

in vivo and in vitro studies. Additionally, the genome encodes a lysozyme (CDS_6), likely enhancing the phage's ability to degrade extracellular polysaccharides, aiding in bacterial lysis and biofilm disruption. Comprehensive scans using CARD and PhageScope did not detect any virulence or antibiotic resistance genes, indicating that qsb1 is a promising and safe candidate for phage therapy. (The NGS raw sequencing data have been deposited in the SRA databases, SRA data accession: PRJNA1170883; The complete genome sequence of the phage has been uploaded to the Bankit database of genbank, accession number: PV107591).

**Phylogenetic analysis.** To elucidate the phylogenetic background of bacteriophage vB_AbaS_qsb1, we performed a BLAST search against all viral genome databases in the NCBI repository, identifying a total of 88 sequences. Among the BLAST results, phages from the genera *Vieuvirus*, *Obolenskvirus*, and unclassified members of the family *Caudoviricetes* were detected.

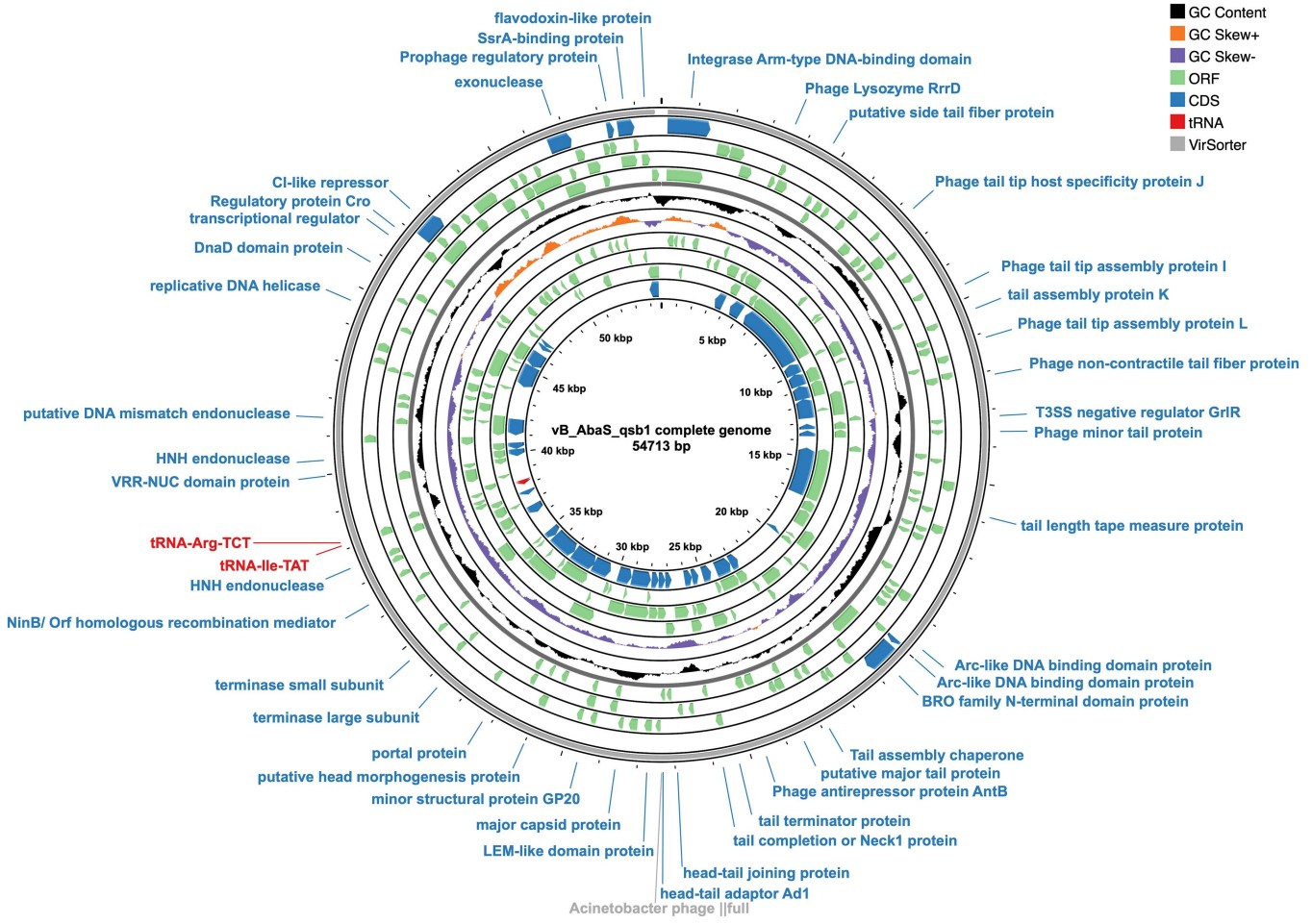

**Fig 2. Whole-genome circular map of bacteriophage vB_AbaS_qsb1.** The black ring represents the GC content, while the orange and purple rings display the CG skew for the positive and negative strands, respectively. Green rings denote predicted open reading frames (ORFs), and blue rings correspond to coding sequences (CDSs) annotated with predicted protein functions. Red labels indicate tRNAs, and grey highlights identifiable viral sequences. Pseudogenes or proteins of unknown function are excluded from the map.

We compared all previously reported complete *Acinetobacter* phage genome sequences (74 in total) and noted that the remaining 14 phages with partial genomes displayed coverage rates ranging from 0% to 49% (S2 Table).The clustering results (Fig 3A) showed that qsb1 was not clustered with the *Vieuvirus* and *Obolenskvirus* genus, which suggests that qsb1 may be significantly different from the phages in these genus. This analysis revealed that vB_AbaS_qsb1 is closely related to *Acinetobacter* phage vB_AbaS_TRS1 and MAG_*Caudoviricetessp*_Isolate_224 (Fig 3A), with both clustering in the same cluster, and indicating genus-level consistency. However, the results of colinearity comparison showed that the genomes of vB_AbaS_qsb1, vB_AbaS_TRS1 and Isolate_224 were extremely different (Fig 3B), and the coverage of vB_AbaS_qsb1 on vB_AbaS_TRS1 and Isolate_224 was extremely low (only 20% and 2%, respectively). The reason for this discrepancy is that the algorithm may classify these individuals that are very different from other species into the same branch (in fact, they have little similarity to each other).Therefore, we suggested that vB_AbaS_qsb1 is a phage independent of any known genus.

Further phylogenetic comparisons were made between qsb1 and representative members of the genera *Vieuvirus* and *Obolenskvirus* in both AAI and GBDP analyses. The AAI values for *Vieuvirus* phages ranged from 58.65% to 73.07%,

**Fig 3. Heat map of complete genome analysis and collinearity analysis. (A)** Heat map of AAI (Average Amino Acid Identity) values and Genome BLAST Distance Phylogeny (GBDP) analysis for 74 complete *Acinetobacter* phage genomes. The AAI values range from 0% (blue) to 100% (red), depicting the degree of genetic similarity among phages. The evolutionary tree on the left and top represents GBDP analysis, showing the phylogenetic relationships among 74 phages. Symbols and colors at the bottom correspond to taxonomic differences at the family, genus, or species level. Vertical

axis labels represent the family, genus, or species, with matching shapes and colors indicating consistency at specific taxonomic levels. **(B)** Collinearity comparison of the vB_AbaS_qsb1 genome with vB_AbaS_TRS1 and Isolate_224. Arrows indicate the location and orientation of CDSs in each genome (right-pointing arrows for the positive strand, left-pointing arrows for the negative strand). The lines between genomes illustrate sequence similarities (as represented in the bottom left corner), and CDSs are color-coded by functional categories, as shown in the legend in the bottom right corner.

while those for Obolenskvirus phages ranged from 32.15% to 62.75% (S3 and S4 Tables). According to ICTV standards, AAI values between 75% and 95% indicate distinct species within the same genus. Thus, the observed AAI and GBDP results support the classification of qsb1 as a distinct species within a novel phage genus.

Similarly, average nucleotide identity (ANI) analysis showed that the OrthoANIu values between qsb1 and representative phages YMC11_11_R3177 and AP22 were only 73.50% and 73.40%, respectively, further supporting its distinction from known species and genera (S5 Table).

The combined AAI, ANI, GBDP, and synteny analyses strongly suggest that qsb1 represents a novel genus, named *Acinibactriovirus*, and a novel species, *Acinibactriovirus lysinus*, within the *Acinetobacter* phage lineage. These results underscore the unique genomic characteristics of qsb1 and open avenues for further taxonomic and functional exploration.

**Phage therapy for acute pulmonary infection caused by *A.baumannii* in mice**

**Establishment of the model of pulmonary infection caused by *A.baumannii* in mice.** A BALB/C mouse model of acute pulmonary infection caused by *A.baumannii* was established by tracheal instillation of bacteria directly into the trachea. Following bacterial infection, mice exhibited clinical signs including hypothermia, reduced activity, and ocular discharge.Over the course of a 7-day observation period, all mice in the $3 \times 10^9$ Colony-forming units (CFU)/mouse group died within 60 hours, and those in the $3 \times 10^8$ CFU/mouse group had a mortality rate of 83.3%. In contrast, only one mouse in the $3 \times 10^7$ CFU/mouse group succumbed (a mortality rate of 16.7%), and no mortality was observed in either the $3 \times 10^6$ CFU/mouse group or the control group (Fig 4A). Based on the median lethal dose ($LD_{50}$) criteria and to ensure the study's comparability, a bacterial dose of $3 \times 10^8$ CFU/mouse was selected for subsequent experiments to construct a reliable model of pulmonary infection in mice.

**The safety of phage therapy.** To assess whether *A.baumannii* phage qsb1 induces severe immune responses or adverse clinical symptoms in mice, phage preparation was instilled into the lungs. Throughout the 7-day observation period, none of the mice in the phage group exhibited any negative clinical symptoms, and no deaths occurred (Fig 4A). Whole blood samples of mice were collected for complete blood count (CBC). The results indicated no significant differences in the levels of white blood cells and neutrophils between the phage group and the control group (Fig 4B). Analysis of relative inflammatory cytokine levels (TNF-α, IL-6, and IL-1β) in mouse lung tissue (Fig 4C) indicated that phage treatment did not trigger a severe immune response in mice. Furthermore, hematoxylin-eosin (HE) staining analysis (Fig 4D) revealed no significant infiltration of inflammatory cells in the lung tissue, no apparent congestion, and the alveolar structure remained intact. Summary, no severe immune responses or lung tissue damage were observed in mice treated with phage qsb1, confirming its safety.

**Phage treatment of mice with pulmonary infection caused by *A.baumannii*.** To investigate the efficacy of different phage doses on the treatment of pneumonia-infected mice ($3 \times 10^8$ CFU/mouse), phage therapy was administered 1 hour post-infection. In the Mock-treated group (saline), a substantial number of mice (4/5) succumbed within 48 hours, resulting in a final survival rate of 20%. Similarly, in both the Medium-dose ($3 \times 10^8$ PFU) and Low-dose ($3 \times 10^7$ PFU) phage treated groups, most mice died (4/5) within 96 hours, with a survival rate of 20% by day 7. The survival rates in these groups were significantly lower than those observed in the control group. Remarkably, in the High-dose ($3 \times 10^9$ PFU) phage treated group, only one mouse died(1/5) following phage treatment, and the survival rate reached 80%, which was not significantly different from that of the control group (Fig 5A).

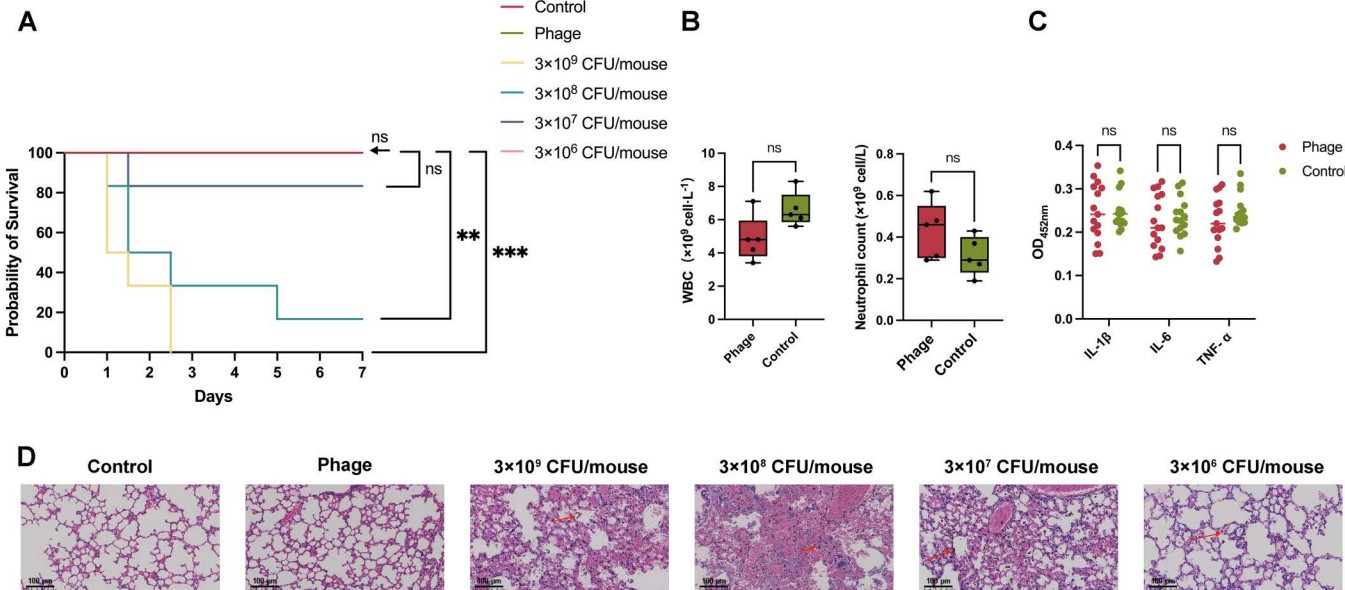

**Fig 4. Mouse model of *A.baumannii* lung infection and phage safety evaluation.** Mice were injected intraperitoneally with cyclophosphamide (CTX). The control group received saline treatment only, while the phage group received $10^9$ PFU of qsb1 only. Other groups were administered varying doses of *A.baumannii* culture. **(A)** Survival curve of mice during the 7-day observation period (n = 6 for the Control, $3 \times 10^9$ CFU/mouse, $3 \times 10^8$ CFU/mouse, $3 \times 10^7$ CFU/mouse and $3 \times 10^6$ CFU/mouse, n = 5 for the Phage group). **(B)** Leukocyte and neutrophil counts in whole blood of the control and phage groups (n = 5). The results indicate no significant differences in leukocyte and neutrophil levels between the phage-treated and control groups. **(C)** Relative levels of inflammatory cytokines (IL-1β, IL-6, and TNF-α) in mouse lung tissues (n = 5). Each experimental group included 5 mice, with each mouse undergoing triplicate measurements. The data demonstrate no significant differences in the levels of IL-1β, IL-6, and TNF-α between the phage group and control groups. **(D)** Pathological examination of mouse lung tissue. Scale bar = 100μm. Statistical analysis of survival curves was performed using the log-rank test, while other data were analyzed using two-way ANOVA. ***$p < 0.001$, **$p < 0.01$, *$p < 0.05$.

After the 7-day observation period, the mice were sacrificed, and blood samples were collected for CBC. The results showed no significant differences in leukocyte and neutrophil counts between the high-dose phage-treated group and the control group (Fig 5B). Histopathological examination of lung tissues with hematoxylin-eosin (HE) staining revealed substantial damage to the alveolar structure in the Mock, Medium-dose, and Low-dose treated groups, including a marked reduction in normal alveoli, inflammatory exudates in the alveolar spaces, thickened and ruptured alveolar septa, and significant congestion and necrosis of lung tissue. In contrast, the high-dose treated group displayed well-preserved lung architecture, with intact alveoli and no evident pathological changes, closely resembling the control group's lung morphology (Fig 5C). In summary, high-dose phage therapy significantly enhances the survival rate of infected mice and effectively prevents the development of lung lesions.

**Dynamic changes in infection status of mice during phage treatment.** As previously established, a phage dose of $3 \times 10^9$ PFU/mouse demonstrated substantial efficacy against lung infection. To further investigate the dynamic progression of the disease during treatment, phages were administered 1 hour post-infection, and 6 mice from each experimental group were euthanized at various time points after infection (12 h, 24 h, 48 h, and 168 h) to collect blood and lung tissue.

Over the 7-day observation period, all mice in the mock-treated group succumbed within 60 hours (Fig 6A). However, 50% of the mice in the phage-treated group survived by the end of the observation period (168 hours). These findings align with our previous studies, demonstrating that phage qsb1 therapy can significantly improve survival rates in mice suffering from acute pneumonia infected by *A.baumannii*. CBC analysis (Fig 6B) showed that phage treatment effectively

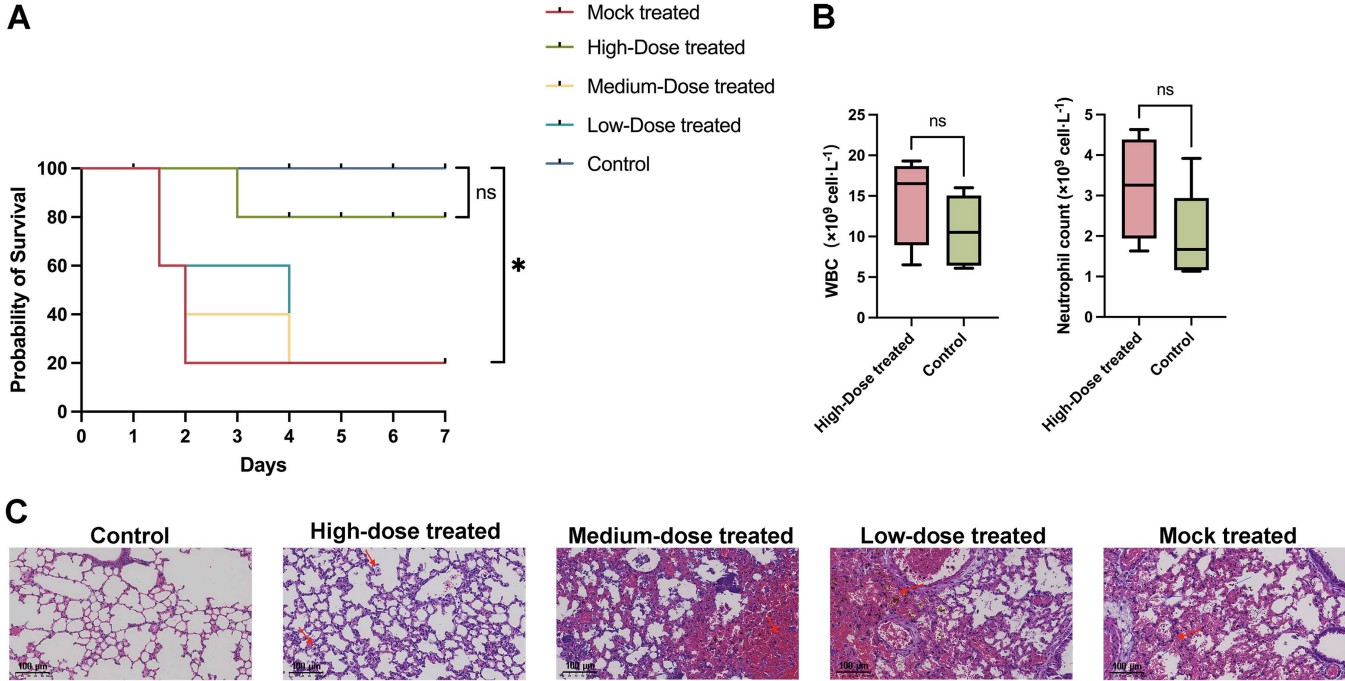

**Fig 5. Phage therapy in lung-infected mice via tracheal instillation.** The control group received only saline without infection, while the Mock-treated group was administered $3 \times 10^8$ CFU of *A.baumannii* ioag01 without any treatment. Mice in the high, medium, and low-dose phage treatment groups were administered $3 \times 10^9$, $3 \times 10^8$, and $3 \times 10^7$ PFU of qsb1, respectively, following infection. **(A)** Survival curve of mice over 7 days following treatment with different doses of phage (n = 5). **(B)** White blood cell and neutrophil counts in mice 7 days post high-dose phage treatment (n = 4). **(C)** Histopathological examination of lung tissue after phage treatment; scale bar = 100 µm. Survival curves were analyzed using the log-rank test, and other data were analyzed using two-way ANOVA. ***$p < 0.001$, **$p < 0.01$, *$p < 0.05$.

suppressed infection-induced inflammation, as evidenced by lower WBC and C-reactive protein (CRP) levels compared to the mock-treated group, indicating reduced systemic immune activation. Besides, the TNF-α levels in the phage-treated group were significantly lower than those in the model control group (Fig 6C). These findings indicate that phage treatment effectively mitigated the inflammatory response in the lungs.

The results of lung homogenate colony counts showed that, The bacterial colonization in the lungs of mice in the mock treatment group remained at a high level ($10^8$ to $10^{10}$ CFU/g) (Fig 6D). In contrast, bacterial colonization in the phage-treated group was significantly lower at all time points. By the 48-hour mark, bacterial colonization in the phage-treated group had decreased to just $10^6$ CFU/g.

Additionally, the phage titer in the lungs of the Phage-treated group was monitored during infection (Fig 6E). At 12 hours after infection, the phage titer was $10^6$ PFU/g. Over time, the phage titer in the lungs of mice in the phage treatment group gradually increased, which was closely associated with the continuous decline in bacterial colonization in the lungs of these mice. By 168 hours, the phage titer had reached $10^{10}$ PFU/g, indicating continuous phage activity in combating infection and without being cleared by the immune system.

Finally, histopathological examination of lung tissue (Fig 6F) revealed that, as the infection progressed, severe lung damage became apparent, including congestion, necrosis, inflammatory cytokine infiltration, and extensive destruction of the alveolar structure. In the Phage-treated group, only minor alveolar edema and limited inflammatory infiltration were observed at all time points, with alveolar structures remaining largely intact, and no signs of congestion, necrosis, or other severe pathological changes.

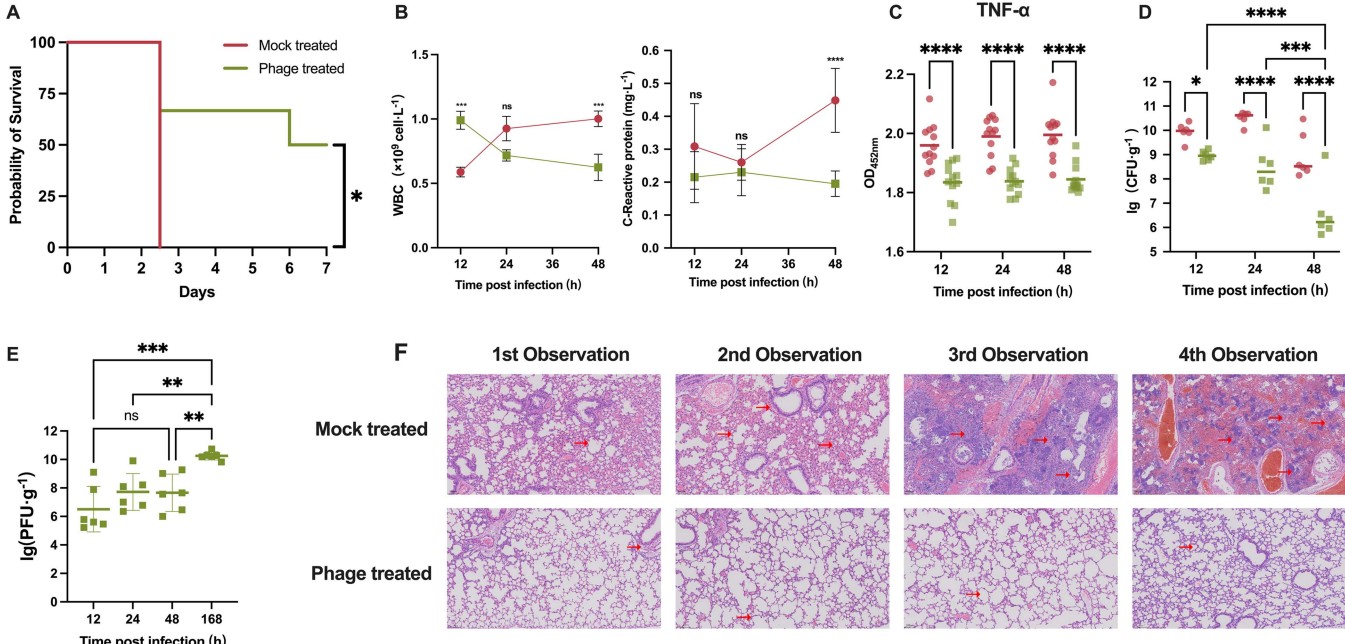

**Fig 6. High-dose phage treatment in mice with *A.baumannii* lung infection.** Saline (red circles) and qsb1 phage (green squares) were administered 1 hour post-infection with *A. baumannii*. The 1st, 2nd, and 3rd observations correspond to 12h, 24h, and 48h, respectively. The 4th observation is at 60h for the Mock treated group (all mice died by 60h) and 128h for the Phage treated group. **(A)** Seven-day survival curve for mice after high-dose phage treatment (n = 6): Mock-treated group (red) and Phage-treated group (green). The survival rate (50%) in the phage-treated group was significantly higher than that in the Mock-treated group at the end of treatment. **(B)** Changes in blood cell counts (WBC and CRP) over the course of treatment (n = 6): Mock-treated group (red) and Phage-treated group (green). A significant increase in white blood cell levels in the Mock-treated group within 48 hours post-infection. In contrast, the Phage-treated group maintained lower white blood cell levels. Similarly, C-reactive protein (CRP) levels in the Mock-treated group increased markedly, whereas those in the Phage-treated group remained stable. **(C)** TNF-α levels in mouse lungs; n = 6 mice per group per time point, with two replicates per mouse. Mock-treated group (red) and Phage-treated group (green). Lines represent the mean. Significant differences in TNF-α levels were observed between the Phage-treated and Mock-treated groups at all time points. **(D)** Bacterial CFU in the lungs of mice; each symbol represents an individual mouse, and lines represent the mean. Results are combination of 1 independent experiment, n = 6 mice per group, normalized to lung weight. The CFU in the lungs of the phage-treated group showed significant or highly significant reductions at all time points compared with the Mock-treated group. **(E)** PFU in the lungs of phage-treated mice. Each symbol represents a logarithmic value, and lines represent the mean. Results represent pooled data from independent experiments, with n = 6 mice per time point, normalized to lung weight. Significant differences in PFU in the lungs were observed at the end of mouse treatment (168 h) and at other time points (12 h, 24 h, and 48 h). **(F)** Histopathological analysis of lung tissue; scale bar = 100 μm. Survival curves were analyzed using the log-rank test, while other data were analyzed using two-way ANOVA. ***$p < 0.001$, **$p < 0.01$, *$p < 0.05$.

## Discussion

The global rise of antimicrobial resistance (AMR) poses a serious public health threat [23,24]. Phage therapy, recognized for its antibacterial properties [11], offers distinct advantages over conventional treatments, including targeted activity, safety, and without disturbing the commensal microbiota [25]. Recent high-profile clinical cases and ongoing trials in countries like the U.S., France, and Sweden underscore its potential in combating multidrug-resistant (MDR) pathogens [26].

In this study, we isolated and characterized the bacteriophage vB_AbaS_qsb1, a virulent phage with potent lytic activity against *A.baumannii*. vB_AbaS_qsb1 demonstrated remarkable stability across a broad temperature (10–60°C) and pH range (pH 3–11), indicating its potential for use under diverse physiological conditions. The phage's short latent period allows rapid bacterial lysis, despite its moderate burst size. These findings will support vB_AbaS_qsb1 as a promising candidate for therapeutic infection of *A.baumannii*.

Genomic analysis of vB_AbaS_qsb1 classified it within the Caudoviricetes family, supporting its designation as the founding member of a novel genus, which we propose to name *Acinibactriovirus*, with *Acinibactriovirus lysinus* as the type

species. Although vB_AbaS_qsb1 harbors an integrase gene (CDS_1) often associated with lysogeny, both experimental data and lifestyle prediction analyses confirmed that it exhibits characteristics of a virulent phage.

The genome of vB_AbaS_qsb1 also encodes a lysozyme (CDS_6), likely enhancing its capability to degrade extracellular polysaccharides, positioning this phage as a potential anti-biofilm and anti-infective tool [27,28]. Intriguingly, qsb1 produced a prominent depolymerase halo during bacterial lysis, though no specific depolymerase gene was detected, suggesting that its depolymerase function may arise from "viral dark matter"—genes that remain uncharacterized.

Safety is a critical factor in the clinical use of phage therapy [27]. In this study, intratracheal administration of $10^9$ PFU vB_AbaS_qsb1 to mice did not induce adverse clinical symptoms, with no increase of inflammatory marker level and no significant lung tissue damage. These results confirm the safety of vB_AbaS_qsb1, essential for its clinical advancement. Moreover, further in vivo experiments demonstrated its therapeutic efficacy, phage significantly reducing bacterial loads of lung and improving survival rates in mice with *A.baumannii* lung infections. The phages persisted in the lungs for up to a week post-treatment without immune system clearance, continuously reducing bacterial load while proliferating. This, combined with their host specificity and minimal impact on beneficial microbiota, highlights their potential as an effective alternative against multidrug-resistant bacterial infections.

While our results are promising, several limitations remain. The efficacy of vB_AbaS_qsb1 in chronic infections hosts warrants further study. Additionally, There was some variability in the survival rate of mice at the endpoint observed in two independent experiments (80% and 50%, respectively), which suggests that the results are not completely reproducible. We speculate that this may be due to potential differences in mouse sensitivity and bacterial virulence between experiments. This also may be attributed to mechanical damage during phage administration or the empirical nature of current phage therapy protocols [29,30]. Therefore, future studies can further evaluate the impact of different administration routes of phage on the survival rate of mice while ensuring safety. Finally, in addition to the route of administration, it is worth further study on whether to use a single phage or a phage cocktail, phage dosage and endotoxin levels in phage preparations, frequency and time of administration, and phage neutralizing antibodies.

In conclusion, We identified and characterized a novel virulent *A.baumannii* phage, vB_AbaS_qsb1, demonstrating strong lytic activity in vitro and in vivo. It effectively reduced bacterial load, improved survival rates, and maintained high lung titers for seven days post-administration, suggesting potential preventive applications. The phage exhibited a favorable safety profile, with no adverse effects observed. Given the rising antibiotic resistance, qsb1 represents a promising therapeutic alternative or adjunct to traditional antimicrobial treatments.

## Materials and methods

### Ethics statement

All protocols in this study were approved by the Committee on the Ethics of Animal Experiments of College of Biological Science and Engineering, Fuzhou University (the authority of the project licence: KS2023-SG-042).

### Bacterial strains and culture conditions

*A.baumannii* ioag01 was obtained from our laboratory in a previous study [31]. In this study, ioag01 was used as a host strain to isolate phages. The frozen bacteria were first streaked on Luria-Bertani Broth for 12 h (37 °C), and then single colonies were picked and enriched in liquid culture medium in a constant temperature shaking incubator at 220 rpm (37 °C, 12 h). The bacterial culture was stored at -80 °C for long-term with 10% glycerol as a cryoprotectant.

### Phage isolation and purification

Water samples were collected from the lake in Qishan Lake Park, Fuzhou City, Fujian Province, China. The samples were centrifuged and filtered through a 0.22 μm PES membrane filter. The filtrate was collected in a sterile centrifuge tube. For

the enrichment of bacteriophages, the bacterial culture (*A. baumannii* ioag01) was first centrifuged and the supernatant was removed. The bacterial precipitate was resuspended in the 2×LB liquid medium and an equal volume of filtrate was added to it. The mixture was co-cultured at 220 rpm (37 °C, 5~7 d). The mixture solution was then centrifuged at 20,000 g (4 °C, 10 min). The supernatant was collected and the presence of phages was detected by the drop method [32]. For positive samples detected by the drop method, the phage suspension was first serially diluted using SM buffer (1.972 g/L $MgSO_4 \cdot 7H_2O$, 5.85 g/L NaCl, 50 mL/L 1M Tris-HCl (pH 7.5), 5.0 mL/L 2% Gelatin). Subsequently, 500 µL of the phage dilution was mixed with 500 µL of bacterial culture and incubated at 37 °C for 10 minutes. The 1 mL mixture was then added to 0.6% semi-solid medium and layered onto the surface of a 1.5% solid medium to form double-layer plates, which were incubated overnight at 37 °C. Individual plaques were picked into SM buffer on the second day. To purify the isolated phage, an individual plaque was picked and passaged at 3~5 times by the double-layer agar (DLA) method. Single plaques were collected to complete the purification and determine the phage titer.

## Phage amplification, concentration, and storage

A 500 µL aliquot of the purified phage lysate was mixed with 500 µL of bacterial culture and incubated overnight at 37 °C. The mixture was then centrifuged at 20,000 g (4 °C, 10 min). The phage-containing supernatant was collected and added to 5 mL of bacterial culture. This mixture was incubated at 220 rpm (37 °C) until the culture became clear. Subsequently, the culture was centrifuged again at 20,000 g (4 °C, 10 min). The supernatant was filtered through a 0.22 µm PES membrane to obtain the phage suspension. The phage suspension was then mixed with a 4× virus concentration solution (400 g/L PEG-8000, 70 g/L NaCl, 100 mL/L 10×PBS buffer, pH 6.8) at a 3:1 (v/v) ratio and allowed to stand at 4 °C overnight. Following this, the mixture was centrifuged at 12,000 g (4 °C, 15 min), and the supernatant was discarded. The phage pellet was resuspended in SM buffer. Finally, the phage particles were stored at -80 °C using 10 mg/mL PEG-4000 as a cryoprotectant for long-term preservation.

## Transmission electron microscopy (TEM)

A 20 µL aliquot of the concentrated phage particle suspension was applied to the surface of a carbon-coated copper grid and allowed to stand for 10 minutes. Excess liquid was then removed using filter paper. The grid was subsequently stained with 20 µL of 2% phosphotungstic acid for 90 seconds to achieve negative staining. After removing the excess stain with filter paper, the grid was left to dry overnight. Imaging was performed using a transmission electron microscope (TEM, HT7800, HITACHI) operating at an accelerating voltage of 100 kV.

## Multiplicity of infection (MOI) of phages

The concentration of the host bacterial culture was determined by turbidimetry, and the bacterial concentration was adjusted to $10^8$ CFU/mL. The titer of phage qsb1 was then determined using the DLA method. Phage and host bacterial cultures were mixed in equal volumes at MOI of 0.01, 0.1, 1, 10, and 100, and incubated with shaking at 220 rpm (37 °C, 6 h). Following incubation, the cultures were centrifuged, and the phage titer was measured. Three independent experiments were performed.

## One-step growth curve of phages

The phage and host bacterial culture were mixed at the optimal multiplicity of infection (MOI = 0.01) determined previously, and incubated at 37 °C for 10 minutes. The supernatant was then discarded, and the precipitate was washed once with LB liquid medium by centrifugation at 20,000 g (4 °C, 1 min). The precipitate was resuspended in LB medium and incubated with shaking at 220 rpm (37 °C, 160 min). Samples were collected every 5 minutes during the first 20 minutes, and every 20 minutes during the final 140 minutes, for phage titer determination. Three independent experiments were performed.

**Temperature and pH stability of phages**

To assess the temperature stability of phage qsb1, a phage suspension with a previously measured titer was incubated at temperatures of 10 °C, 20 °C, 40 °C, 60 °C, 80 °C, and 100 °C for 30, 60, and 90 minutes, respectively, followed by phage titer determination. Three independent experiments were performed.

Similarly, to evaluate the pH stability of phage qsb1, 100 μL of phage suspension (with a known titer) was mixed with 900 μL of SM buffer at various pH levels (1, 3, 5, 7, 8, 9, 11, 13) and incubated (4 °C, 1 h). Subsequently, the pH of the mixtures was adjusted with appropriate amounts of NaOH or HCl to a level suitable for bacterial growth, and the phage titer was measured. Three independent experiments were performed.

**Extraction, sequencing, and analysis of phage genomes**

The total genome of the phage was extracted using the phenol-chloroform method, with nuclease digestion performed beforehand to eliminate host bacterial nucleic acids from the sample. The concentration of the extracted genome was measured using a Nanodrop 2000 spectrophotometer (Thermo Fisher Scientific). The genome was treated with RNase A (Gproan, GPE007001), DNase I (Thermo Fisher Scientific, EN0521), and S1 nucleases (Thermo Fisher Scientific, EN0321) to confirm its nucleic acid type and structural integrity. Subsequently, Pulsed-Field Gel Electrophoresis (Pippin-pluse, sage science) was employed to confirm the size of the phage genome nucleic acid.Next-generation sequencing (NGS) library of the phage genome was constructed using the Nextera library preparation kit, followed by high-throughput sequencing on the Illumina NovaSeq X Plus system.

Post-sequencing, quality control of the raw data was conducted to ensure high-quality reads, which were then assembled using MEGAHIT software [33]. For functional annotation of the assembled phage sequences, several bioinformatics tools were utilized, including Prokka [34], RAST [35], GeneMarks [36], CARD [37], PhageScope [38],and Phastest [39]. Phage lifestyle prediction was conducted using the DeepPhage [40] tool to determine whether the isolated phage exhibits a lytic or lysogenic lifecycle.The integrated gene annotation results provided comprehensive genomic information for the phage qsb1.Genomic features were visualized using the CGView [41] online server.

**Phylogenetic analysis**

The phage genome was compared with the virus database in the NCBI database by BLAST (using the nt_Viruses datebase and algorithm of Highly similar sequences (megablast)), and then the genomes of *Acinetobacter* phages with full genome sequences in the comparison results were downloaded (74 of 88 sequences) for phylogenetic tree construction. Phylogenetic trees and clustering of phage genomes were constructed using VICTOR [42],which based on a Genome BLAST Distance Phylogeny (GBDP) to obtain a more accurate taxonomic classification. Average nucleotide identity (ANI) analysis was performed using the ANI Calculator [43]. Average amino acid identity (AAI) analysis was carried out based on a pairwise comparison of *Acinetobacter* phage sequences using CompareM (https://github.com/dparks1134/CompareM) with default parameters. For comparative analysis of whole-genome collinearity among phages, RStudio [44] with the gggenomes package [45] was employed to generate a whole-genome comparison visualization map.

**Mice**

For all mouse experiments, we selected male BALB/C mice aged 6–8 weeks. Upon arrival at the facility, the mice randomly were housed in clean individually ventilated cages (IVC) with 5–6 mice per group, Random numbers were generated using the standard = RAND() function in Microsoft Excel. All mice were used after a 7-day acclimatization period. They were fed with irradiated sterilized feed and provided distilled water sterilized by high-pressure wet heat. The housing conditions were maintained at a constant temperature of 24–26 °C, humidity of 50%-60%, with a 12-hour light/dark cycle.

## Complete blood count

After collecting whole blood from the mice, blood parameters were analyzed within 1 hour using an automated blood cell counter (XN-550, Sysmex America).

## Preparation of bacterial culture and phage lysates

The bacterial culture of *A.baumannii* was prepared as described above, with the concentration adjusted by turbidimetry. A 1 mL aliquot of the bacterial culture was centrifuged at 20,000 g (4 °C, 10 min), and the pellet was collected. The pellet was washed with 0.9% NaCl, and then resuspended in 30 µL of 0.9% NaCl for use. Bacteria were removed from the phage lysates using a 0.22 µm microporous membrane, and the phage titer was subsequently determined using the DLA method. Subsequently, the phage suspension was centrifuged at 100,000 g (4 °C, 2 h). The phage pellet was resuspended in 0.9% NaCl to a final concentration of $1 \times 10^{12}$ PFU/mL for further use.

## Establishment of a mouse pneumonia model via intratracheal instillation

In previous studies, tracheal access often involved surgical procedures to locate the trachea for drug administration or model establishment, which could introduce experimental variability, such as post-surgical infections or complications during the procedure. To mitigate these risks, we employed a Non-invasive intratracheal instillation approach to construct the mouse pneumonia model [46]. This method is straightforward, rapid, and effective. The anesthetized mouse (Mice were anesthetized via intraperitoneal injection of 2.5% Avertin at a dosage of 375 mg/kg) was positioned supine on an intubation platform, with the head secured, and a disposable indwelling needle was inserted into the glottis using a small animal laryngoscope to complete the endotracheal intubation.

## The model of pulmonary infection caused by *A.baumannii* in mice and phage safety

To induce neutropenia and simulate the immunocompromised state of clinical patients, mice were intraperitoneally injected with cyclophosphamide (CTX) at doses of 200 mg/kg and 150 mg/kg, 4 days and 1 day prior to infection, respectively. In previous studies, a bacterial dose ranging from $10^8 \sim 10^9$ CFU was commonly used to establish an acute pulmonary bacterial infection model in mice [47]. In this study, mice were anesthetized and inoculated via tracheal instillation with varying doses of *A.baumannii* ioag01 ($3 \times 10^9$ CFU/mouse, $3 \times 10^8$ CFU/mouse, $3 \times 10^7$ CFU/mouse, $3 \times 10^6$ CFU/mouse). In the phage-treated group, mice were administered $1 \times 10^9$ PFU/mouse of phage qsb1 via tracheal instillation, while the control group received an equivalent volume of 0.9% NaCl. Mice were observed for seven days, and survival rates were recorded to determine the median lethal dose (LD50) of the bacteria and to evaluate the safety of the phage treatment.

## Phage treatment for acute pulmonary infection caused by *A.baumannii* in mice

After determining the appropriate bacterial dose ($3 \times 10^8$ CFU/mouse) for the mouse lung infection model, phages were used to treat the infected mice to assess therapeutic efficacy. Twenty-five BALB/C mice were randomly divided into five groups. They were intraperitoneally injected with 200 mg/kg and 150 mg/kg of CTX, 4 days and 1 day before infection, respectively. Mice were anesthetized and infected with $3 \times 10^8$ CFU of *A.baumannii* ioag01. One hour post-infection, 30 µL of 0.9% NaCl was administered by tracheal instillation (mock-treated group). The experimental groups received varying phage titers ($3 \times 10^9$ PFU, high-dose; $3 \times 10^8$ PFU, medium-dose; and $3 \times 10^7$ PFU, low-dose). The control group received 30 µL of 0.9% NaCl initially, and an additional 30 µL of 0.9% NaCl one hour later to control for procedural variables. All mice were observed for seven days, during which their conditions and mortality were recorded.

## High-dose phage treatment for acute pulmonary infection caused by *A.baumannii* in mice

To further evaluate the pathological changes during infection and the therapeutic effect of high-dose phage treatment, 48 BALB/C mice were randomly divided into two groups. They were injected intraperitoneally with 200 mg/kg and 150 mg/

kg of CTX, 4 days and 1 day prior to infection, respectively. After anesthesia, a lung infection model was established as previously described. One hour after bacterial infection, the experimental group received $3 \times 10^9$ PFU of phage by tracheal instillation, while the control group received 30 µL of 0.9% NaCl. Subsequently, six mice were euthanized by cervical dislocation at 12、24、48 and 168 hours post-infection. Whole blood and lung tissues were collected aseptically from each mouse. Whole blood and lung tissues were collected under sterile conditions. CBC were conducted to evaluate systemic responses, while bacterial and phage colonization in the lungs was assessed by homogenizing the right lung. The left lung was fixed in paraformaldehyde and stained with hematoxylin-eosin (HE) to assess lung pathology.

## Statistical analysis

Data were evaluated using two-way ANOVA or the log-rank test for survival curves, unless otherwise indicated. A p-value of ≤0.05 was considered statistically significant.

## Supporting information

**S1 Fig. Pulsed-field electrophoresis of vB_AbaS_qsb1 genome.** M: DNA Size Standards - 5 kb ladder (BIO-RAD); 1: vB_AbaS_qsb1 genome.
(TIF)

**S2 Fig. Validation of the nucleic acid type of vB_AbaS_qsb1; M: Takara λ-Hind III DNA marker; 1: vB_AbaS_qsb1 genome; 2: DNAse I treatment; 3: RNAse A treatment.**
(TIF)

**S3 Fig. Validation of the nucleic acid structure of vB_AbaS_qsb1; M: Takara λ-Hind III DNA marker; 1: vB_AbaS_qsb1 genome; 2: S1 nuclease treatment.**
(TIF)

**S1 Table. Genome-wide gene annotation information for vB_AbaS_qsb1.**
(XLSX)

**S2 Table. Partial genome alignment results obtained by BLAST.**
(XLSX)

**S3 Table. Mean AAI value of phage of *Vieuvirus genus.***
(XLSX)

**S4 Table. Mean AAI value of phage of *Obolenskvirus genus.***
(XLSX)

**S5 Table. OrthoANIu results against vB_AbaS_qsb1.**
(XLSX)

**S1 Methods. For analysis of bacterial and phage colonization, inflammatory cytokine levels, and histopathological examination in mice.**
(DOCX)

## Acknowledgments

**Declaration of Generative AI and AI-assisted technologies in the writing process:** During the preparation of this work the author(s) used Chat-GPT 4o in order to improve the language and readability, with caution. After using this tool, the authors reviewed and edited the content as needed and take full responsibility for the content of the publication.

## Author contributions

**Conceptualization:** Jun Lin, XiaoLing Yu.

**Data curation:** JiaWang Wang.

**Formal analysis:** JiaWang Wang.

**Funding acquisition:** Huan Hu, Jun Lin, XiaoLing Yu.

**Investigation:** JiaWang Wang, Huan Hu, QingQing Wang, WenQian Jiang, XueMei Hou.

**Methodology:** Jun Lin, XiaoLing Yu.

**Project administration:** XueMei Hou, Jun Lin, XiaoLing Yu.

**Resources:** Jun Lin, XiaoLing Yu.

**Software:** Huan Hu, TianZhu Zhu, XiaoHan Ren.

**Supervision:** XueMei Hou, Jun Lin, XiaoLing Yu.

**Validation:** JiaWang Wang, TianZhu Zhu.

**Visualization:** JiaWang Wang, TianZhu Zhu, XiaoHan Ren.

**Writing – original draft:** JiaWang Wang.

**Writing – review & editing:** XueMei Hou, Jun Lin, XiaoLing Yu.

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
