## [Decision Letter · Decision Letter 0]

PPATHOGENS-D-24-02438

A Novel Genus of Virulent Phage Targeting Acinetobacter baumannii: Efficacy and Safety in a Murine Model of Pulmonary Infection

PLOS Pathogens

Dear Dr. Yu,

Thank you for submitting your manuscript to PLOS Pathogens. After careful consideration, we feel that it has merit but does not fully meet PLOS Pathogens's publication criteria as it currently stands. Therefore, we invite you to submit a revised version of the manuscript that addresses the points raised during the review process.

Please submit your revised manuscript within 60 days Mar 24 2025 11:59PM. If you will need more time than this to complete your revisions, please reply to this message or contact the journal office at plospathogens@plos.org. Please include the following items when submitting your revised manuscript:

We look forward to receiving your revised manuscript.

Kind regards,

Jose Luis Balcazar, Ph.D.

Academic Editor

PLOS Pathogens

Robert Kalejta

Section Editor

PLOS Pathogens Sumita Bhaduri-McIntosh

Editor-in-Chief

PLOS Pathogens

orcid.org/0000-0003-2946-9497

Michael Malim

Editor-in-Chief

PLOS Pathogens

orcid.org/0000-0002-7699-2064

**Journal Requirements:**

1)  We do not publish any copyright or trademark symbols that usually accompany proprietary names, eg ©,  ®, or TM  (e.g. next to drug or reagent names). Therefore please remove all instances of trademark/copyright symbols throughout the text, including:

- TM on page: 26 line 502.

2) We have noticed that you have uploaded Supporting Information files, but you have not included a list of legends. Please add a full list of legends for your Supporting Information files after the references list.

3) Some material included in your submission may be copyrighted. According to PLOSu2019s copyright policy, authors who use figures or other material (e.g., graphics, clipart, maps) from another author or copyright holder must demonstrate or obtain permission to publish this material under the Creative Commons Attribution 4.0 International (CC BY 4.0) License used by PLOS journals. Please closely review the details of PLOSu2019s copyright requirements here: PLOS Licenses and Copyright. If you need to request permissions from a copyright holder, you may use PLOS's Copyright Content Permission form.

Potential Copyright Issues:

i) Please confirm (a) that you are the photographer of 1A, or (b) provide written permission from the photographer to publish the photo(s) under our CC BY 4.0 license.

ii) Graphical Abstract. Please confirm whether you drew the images / clip-art within the figure panels by hand. If you did not draw the images, please provide (a) a link to the source of the images or icons and their license / terms of use; or (b) written permission from the copyright holder to publish the images or icons under our CC BY 4.0 license. Alternatively, you may replace the images with open source alternatives. See these open source resources you may use to replace images / clip-art:

4) In the online submission form, you indicated that "The datasets generated during and/or analyzed during the current study are available from the corresponding author on reasonable request." All PLOS journals now require all data underlying the findings described in their manuscript to be freely available to other researchers, either

1. In a public repository

2. Within the manuscript itself

3. Uploaded as supplementary information.

2) State what role the funders took in the study. If the funders had no role in your study, please state: "The funders had no role in study design, data collection and analysis, decision to publish, or preparation of the manuscript.".

**Reviewers' Comments:**

Reviewer's Responses to Questions

**Part I - Summary**

Reviewer #1: The paper presents a proposal for using the lytic bacteriophage vB_AbaS_qsb1, which specifically lyses Acinetobacter baumannii. I believe that the paper requires some minor and major corrections before it can be accepted.

Reviewer #2: This study by Wang and colleagues relates the characterization of a novel phage and its assessment for treating Acinetobacter baumannii pulmonary infection in mice. On the phage characterization section, the one-step growth does not seems to be appropriately executed, but it does not jeopardize the rest of the manuscript. The in vivo data could be presented in greater details and discussion.

Here are more specific comment/questions

L24 and in other places in the manuscript: it is unclear if these authors have contacted the ICTV to request the classification of their phage into a new genus. Therefore, they must be more cautious in the naming convention. Using the phrasing “…we propose to name Acinibactriovirus…” instead of “we named”. I strongly encourage the authors to contact the ICTV committee to formulate their proposition of a new genus.

L37: replace unique by novel

L59: please define the term ICU

L68-69: this sentence is unclear without the word “antibiotics” added after “inherent” and after “acquire”. Please rephrase

L86-89: please simplify by making two sentences. 1) .. we isolated from sewage a novel… baumannii. 2) We characterized its biological properties and genomic features.

L93: you should not jump into conclusions before introducing data. The significant potential should be removed from this sentence.

L102: remove “a novo-“

L104: Remove “potent”.

You should NOT give a full name to this phage before describing the genome that supports the naming convention. Instead use qsb1 here and later mention that the full name is vB…. But that in the manuscript you’ll keep qsb1.

L106: Please rephrase as “following preprepared” is quite an unusual term.

L111: correct the typo to the word “dimensions”.

L116: typing GMDCC 64831-B1 in Google does not provide a way to access this resource. The genomic sequence of this phage should be deposited in Genbank or equivalent public database.

L118 and figure 1C: this graph represents the total amount of particles produced under different starting conditions that are not detailed in the legend. The initial amount of phage should be indicated and the ratio between initial and the final amount of particles should be a better representation of the “replication efficiency”. The individual points of the three replicates should be indicated as in panel F. In panel E as well the three independent values should be presented.

L121: the one-step growth has not been performed in the optimal conditions. Usually, after mixing phage and bacteria at a low MOI, the suspension is diluted to prevent re-infection of bacteria at the end of the first infection cycle. Here, the latent period, which should be represented by a flat line with a stable amount of phage over time is in fact characterized by an increasing amount of phages, which suggest that some bacteria are already producing new particles. It is therefore likely that the calculation of the burst size is compromised. Authors should also provide a reference to the protocol and calculation used to determine these parameters. Finally, in panel F, there is no mention in the legend of the statistical test used.

In the legend of Fig S2, replace “nuclear” by “nucleic acid”

L150-152: since 95 CDS were detected and that 42 have a known function, then why only 51 instead of 53 have no function (replace role by function)?

L177-188: this paragraph should be revised as the data provided on figure 3A is quite unusual. How could it be possible that isolate 224 and TRS1 cluster together with qsb1 while they barely share 2 and 20 % similarity. Then how the authors decided to next focus on Vieuvirus… this is clearly unclear!

L201: it is now more clear on which basis the authors suggest to name a new genus.

L223: here you should mention that your mice are neutropenic as they received cyclophosphamide, as mentioned in the method section. Therefore, it is unclear if the authors have evaluated phage safety on neutropenic mice?

In figure 4, panel A, the group of phage treated is invisible. I also suggest to replace “phage treated” by “phage”. The word treated is implying that mice may be infected.

Panel B, the individual 5 values should be indicated for WBC and neutrophils counts. The associated methods should be in the main manuscript and not in the supplement.

Panel C: it seems that authors have included technical replicates, which is fine as long as it is mentioned in the legend.

L265-267: Please add the infection dose of bacteria (… infected mice (3x10^8 CFU) as well as the phage dose corresponding to the low/medium/high to help the reader appreciate the results without looking for the legend of figure 5.

Panel B: why the values are not shown for the 3 other groups (medium/low and mock)?

L307-329: I suggest the authors to reshuffle the organization of the data by presenting the dat from panels ACDBEF.

L307-309: I disagree with the term “continuous rise” as the main rise is observed between 12 and 24h and that almost no rise is observed between 24 and 48h. The second graph on CRP in Panel B is not commented and the legend is inappropriate.

L319-320: this sentence is unclear and should be rephrased

L341: please indicate how many independent experiments were pooled, or are authors referring to independent groups for each time point ? It these experiments are really independent, then they should have 3 independent groups of mock-treated, which does not seem to be the case. Please clarify.

L350: the lines representing the mean are barely visible, make them thicker

L374-375: this sentence does not add any value to the discussion

L382-384: this is a redundant statement, please remove it.

L387: the term “stable inflammatory markers” is confusing, please consider using “no increase of inflammatory marker level”.

L398: the authors focused on the data from fig 6 to state that survival is about 50%. However, I propose them to critically re-assess their survival data by analyzing them at the same time point, ie 60H. At this time point, in fig 4, the dose of 3x10^8 CFU leads to roughly 50% of mortality, while it rises to 80% in figure 5 and 100% in figure 6. It is clearly suggesting that either mice were more susceptible or the exact infection dose was strictly identical (not taking into consideration that tracheal instillation is quite challenging). Nevertheless, in figure 5 and 6, the mortality of the phage treated group (10^9 PFU) leads to 0% and 35%, respectively. By comparing the difference between mock vs phage-treated groups, in figure 5 the gain in survival is 80% and 65%, respectively, which means an average “gain in survival” of approx. 70%.

As a reviewer I’m not asking the authors to adhere to this suggestion but rather critically discuss their data that show that the initial conditions were not so reproducible, which are therefore blurring the precise evaluation of the efficacy of the phage treatment.

L406-415: most of this paragraph is redundant with other elements, it could be easily streamlined.

L535: What do you mean by “section 2.1”?

L539: replace “suspensions” by “lysates”

L549: please indicate the anesthesia procedure and details

L555: CTX should be mentioned much earlier. Was any control or reference of the neutropenia achieved by this protocol?

L612: the DOI link to the original data is unclear. It provides access to two tables, named Vertical force and Shear force, which are not related to this study. This should be fixed.

**Part II – Major Issues: Key Experiments Required for Acceptance**

Reviewer #1: Please revise the text in general, as several formatting mistakes have been identified (spaces, no italic words, etc.).

Enhance the quality of figures.

Figure 4, 5, and 6. Use an arrow or a line to indicate the damage.

Line 221-231: What are the clinical symptoms that A. baumannii causes in mice?

Why didn’t you evaluate the virulence of the reisolated bacteria to determine if phage therapy reduces its virulence?

Additionally, why didn’t you evaluate the virulence of phage-resistant colonies of A. baumannii?

Reviewer #2: see the above section

**Part III – Minor Issues: Editorial and Data Presentation Modifications**

Reviewer #1: Line 78: in vitro must be in italic format.

Line 86: space after point “].In this…”

Line 94: again space after point.

Line 97: in vivo must be in italic format.

Line 106: preprepared?

Line 110: again space after point.

Line 121: again space between phase(about…

Line 122: again space after point.

Line 331: Text says “Figure 5: High-dose phage treatment in mice with A.baumannii lung infection.” However, is figure 6 instead.

Reviewer #2: see the above section

PLOS authors have the option to publish the peer review history of their article (what does this mean? ). If published, this will include your full peer review and any attached files.

**Do you want your identity to be public for this peer review?** For information about this choice, including consent withdrawal, please see our Privacy Policy .

Reviewer #1: No

Reviewer #2: No

**Figure resubmission:**
---

## [Decision Letter · Decision Letter 1]

PPATHOGENS-D-24-02438R1

A Novel Genus of Virulent Phage Targeting Acinetobacter baumannii: Efficacy and Safety in a Murine Model of Pulmonary Infection

PLOS Pathogens

Dear Dr. Yu,

Thank you for submitting your manuscript to PLOS Pathogens. After careful consideration, we feel that it has merit but does not fully meet PLOS Pathogens's publication criteria as it currently stands. Therefore, we invite you to submit a revised version of the manuscript that addresses the points raised during the review process.

Please submit your revised manuscript within 30 days Jul 15 2025 11:59PM. If you will need more time than this to complete your revisions, please reply to this message or contact the journal office at plospathogens@plos.org. Please include the following items when submitting your revised manuscript:

We look forward to receiving your revised manuscript.

Kind regards,

Jose Luis Balcazar, Ph.D.

Academic Editor

PLOS Pathogens

Robert Kalejta

Section Editor

PLOS Pathogens

Sumita Bhaduri-McIntosh

Editor-in-Chief

PLOS Pathogens

orcid.org/0000-0003-2946-9497

Michael Malim

Editor-in-Chief

PLOS Pathogens

orcid.org/0000-0002-7699-2064

**Additional Editor Comments:**

Although the manuscript has been improved, it still requires minor revisions before it can be accepted for publication. Therefore, it would benefit from being revised according to the suggestions provided.

Additional comments:

The manuscript contains several typographical errors that should be corrected.

The full species name should be used at the beginning of the abstract. For example, “*Acinetobacter baumannii* is a notable opportunistic…” should be used instead of “A. baumannii is a notable opportunistic…”.

Line 57. “*Acinetobacter baumannii* is a Gram-negative…” instead of “A.baumannii is a Gram-negative..”

Lines 167 and 522. “PhageScope” instead of “phagescope”

Line 167. “CARD” should be used instead of "CRAD". Moreover, this database should be properly referenced in the Materials and Methods section.

In line 104 and other instances throughout the manuscript, the strain name should be presented in regular (non-italic) font.

Spacing between words and after punctuation marks should be carefully reviewed. For example, “A. baumannii” should be used instead of “A.baumannii”. 

Line 530. “Acinetobacter” instead of “AcineAtobacter”.

**Journal Requirements:**

1) Please amend your detailed Financial Disclosure statement. This is published with the article. It must therefore be completed in full sentences and contain the exact wording you wish to be published. State what role the funders took in the study. If the funders had no role in your study, please state: "The funders had no role in study design, data collection and analysis, decision to publish, or preparation of the manuscript.".

**Reviewers' Comments:**

Reviewer's Responses to Questions

**Part I - Summary**

Reviewer #3: In this manuscript, the authors isolated and characterized a virulent phage, vB_AbaS_qsb1, capable of specifically lysing Acinetobacter baumannii. They identified that this phage belongs to a newly proposed genus, Acinibactriovirus. The isolated phage exhibited stability under various temperature and pH conditions and did not contain any genes associated with virulence or antibiotic resistance. In safety assays, vB_AbaS_qsb1 caused no adverse effects in mice. Furthermore, in therapeutic experiments, it demonstrated significant protection against A. baumannii-induced pneumonia, reducing both bacterial load and inflammatory markers. These findings highlight the potential of vB_AbaS_qsb1 as a promising option for phage therapy against antibiotic-resistant infections.

**Part II – Major Issues: Key Experiments Required for Acceptance**

Reviewer #3: (No Response)

**Part III – Minor Issues: Editorial and Data Presentation Modifications**

Reviewer #3: The reviewers’ comments have been addressed, resulting in an improved manuscript. Therefore, I consider it suitable for publication. However, I would like to point out a few minor comments:

L59: "Intensive Care Unit" appears in a different font compared to the rest of the text.

L87: Acinetobacter baumannii is not italicized. Additionally, since the full name of the bacterium has already been mentioned earlier, it should be abbreviated as A. baumannii.

L167: The absence of virulence genes and ARGs in the genome is assessed using the CRAD and Phagescope databases. Is “CRAD” correct, or did the authors mean “CARD”? Moreover, this database should also be mentioned in the Methods section.

PLOS authors have the option to publish the peer review history of their article (what does this mean? ). If published, this will include your full peer review and any attached files.

**Do you want your identity to be public for this peer review?** For information about this choice, including consent withdrawal, please see our Privacy Policy .

Reviewer #3: No

**Figure resubmission:**
---

## [Editor Report · Decision Letter 2]

Dear Ph.D. Yu,

We are pleased to inform you that your manuscript 'A Novel Genus of Virulent Phage Targeting Acinetobacter baumannii: Efficacy and Safety in a Murine Model of Pulmonary Infection' has been provisionally accepted for publication in PLOS Pathogens.

Best regards,

Jose Luis Balcazar, Ph.D.

Academic Editor

PLOS Pathogens

Robert Kalejta

Section Editor

PLOS Pathogens

Sumita Bhaduri-McIntosh

Editor-in-Chief

PLOS Pathogens

orcid.org/0000-0003-2946-9497

Michael Malim

Editor-in-Chief

PLOS Pathogens

orcid.org/0000-0002-7699-2064

All reviewer comments have been carefully considered and addressed. Many thanks!
---

## [Editor Report · Acceptance letter]

Dear Ph.D. Yu,

We are delighted to inform you that your manuscript, "A Novel Genus of Virulent Phage Targeting Acinetobacter baumannii: Efficacy and Safety in a Murine Model of Pulmonary Infection," has been formally accepted for publication in PLOS Pathogens.

Best regards,

Sumita Bhaduri-McIntosh

Editor-in-Chief

PLOS Pathogens

orcid.org/0000-0003-2946-9497

Michael Malim

Editor-in-Chief

PLOS Pathogens

orcid.org/0000-0002-7699-2064